# Genetics of Germination and Seedling Traits under Drought Stress in a MAGIC Population of Maize

**DOI:** 10.3390/plants10091786

**Published:** 2021-08-27

**Authors:** Soumeya Rida, Oula Maafi, Ana López-Malvar, Pedro Revilla, Meriem Riache, Abderahmane Djemel

**Affiliations:** 1Higher National Agronomic School (ENSA), L-RGB, Hassan Badi, El Harrach, Algiers 16004, Algeria; soum.rida@gmail.com (S.R.); oulamaafi@gmail.com (O.M.); m.riache93@gmail.com (M.R.); djemeldahmane@yahoo.fr (A.D.); 2Facultad de Biología, Departamento de Biología Vegetal y Ciencias del Suelo, Agrobiología Ambiental, Calidad de Suelos y Plantas, Universidad de Vigo, As Lagoas Marcosende, 36310 Vigo, Spain; 3Misión Biológica de Galicia (CSIC), Apartado 28, E-36080 Pontevedra, Spain; previlla@mbg.csic.es

**Keywords:** maize, drought stress, genome wide association study (GWAS), germination, seedlings, candidate genes

## Abstract

Drought is one of the most detrimental abiotic stresses hampering seed germination, development, and productivity. Maize is more sensitive to drought than other cereals, especially at seedling stage. Our objective was to study genetic regulation of drought tolerance at germination and during seedling growth in maize. We evaluated 420 RIL with their parents from a multi-parent advanced generation inter-cross (MAGIC) population with PEG-induced drought at germination and seedling establishment. A genome-wide association study (GWAS) was carried out to identify genomic regions associated with drought tolerance. GWAS identified 28 and 16 SNPs significantly associated with germination and seedling traits under stress and well-watered conditions, respectively. Among the SNPs detected, two SNPs had significant associations with several traits with high positive correlations, suggesting a pleiotropic genetic control. Other SNPs were located in regions that harbored major QTLs in previous studies, and co-located with QTLs for cold tolerance previously published for this MAGIC population. The genomic regions comprised several candidate genes related to stresses and plant development. These included numerous drought-responsive genes and transcription factors implicated in germination, seedling traits, and drought tolerance. The current analyses provide information and tools for subsequent studies and breeding programs for improving drought tolerance.

## 1. Introduction

Predictions of climate change leading to abiotic stresses, such as rising drought events and higher temperature, indicate an intensification in their severity and frequency [1], and an increased variability of rainfall, in the next 40 years [2]. Drought is one of the main challenges of agriculture at the global scale because it is the most serious and prevalent abiotic stress hampering the potential productivity of crops [3]. Most crops are susceptible to drought and can experience yield losses of more than 50%. Moreover, maize is more sensitive to drought than other cereals [4]. As a result, annual economic losses due to the reduction of maize production caused by drought stress amount to 15–20% [4,5,6,7].

The global demand for maize is increasing as a source of food, feed, and industrial material in line with the increase in the size of the human population [8]. However, future drought stress caused by accelerated global warming will be detrimental for maize production [9]. A share in excess of 20% of the maize growing area will be affected by drought, particularly in the East Asian, South American, and Western European major maize producing zone [10,11]. Considering the agricultural and economic importance of maize, and the constraints imposed by current climate change, the improvement of maize for drought tolerance is a priority in breeding programs as an important part of the solution to stabilize global maize production [12].

Because drought tolerance involves diverse physiological parameters and is controlled by a large number of genes with small effects, its improvement is difficult [13]. Therefore, identification of the genetic regulation of drought tolerance is of great importance for establishing efficient breeding programs [14]. Genetic mapping is a powerful strategy that exploits genomic information to dissect complex traits and identifies genetic determinants that may lead to crop improvement [15]. A higher resolution is possible through association mapping than with traditional QTL mapping, and association mapping is suitable for simultaneously mapping multiple traits [16]. Furthermore, maize is an ideal model plant to conduct association studies because of its great genetic diversity and genome-wide rapid linkage disequilibrium (LD) decay [17].

The genome-wide association study (GWAS) is a widely adopted technique for decoding genotype–phenotype associations in many species through advances in Next Generation Sequencing technologies [18]. In maize, GWAS has been used to locate regions of the genome associated with complex traits [19,20,21] and contributed to the understanding of maize functional genomics and genetics [18]. GWAS has been widely used in maize as an effective means to find candidate genes. Tian et al. [22] determined the genetic basis of important leaf architecture traits and identified some of the key genes. Revilla et al. [23] conducted GWAS in temperate maize inbred lines to identify genes related to cold tolerance. Wang et al. [24] performed GWAS and identified 42 candidate genes that contribute to 55.2% of the natural variation of drought tolerance in maize seedlings. Furthermore, several GWAS reports are available on other target genes for crop improvement [25]. The inclusion of more diverse founders is one means to further increase GWAS mapping resolution [18].

The multi-parent advanced generation inter-cross (MAGIC) population represents a new and powerful tool allowing the high-resolution mapping of quantitative traits [26,27]. Crop improvement against abiotic stresses such as drought requires large germplasm diversity screening [27], and the combination of multiple founders provides a higher genetic and phenotypic diversity within a single mapping population. Thus, the MAGIC population is a useful tool to develop strategies to cope with drought stress.

Water scarcity influences crops at different levels of their growth and development [28], and affects maize from germination to maturity [29]. In addition to yield losses, drought limits the germination rate [30] and reduces seedling establishment [31]. Although drought tolerance at different stages of development is unrelated [32], several studies and breeding programs have been based mainly on yield, flowering, and adult vegetative phases. In contrast, limited research attention has been paid to study maize at early stages of development.

Evaluation of drought tolerance at the seedling stage is necessary to predict good crop stands at maturity. In the early stages of cultivation, drought causes a significant decline in germination rates and increase in seedling mortality, and is detrimental to plant viability and development [33]. Phenotyping young seedlings under controlled conditions is a useful approach to identify candidate drought-tolerant genotypes, and can reduce the laborious and time-consuming selection under field conditions [34]. Drought stress reduces the phenotypic expression of all the seedling traits [35]. Shoot and root traits are potentially used in drought-tolerance screenings, and are inhibited by drought [36,37].

Because the assessment of root behavior is not possible in the field [38], several authors have reported that the induction of water stress by Polyethylene glycol (PEG) is a reliable approach for the assessment of water stress in germplasm collections of maize [39,40,41]. This approach has been used for the study of water stress at early stages [41,42,43,44,45]. Direct studies on root traits are highly complex. Thus, as an alternative, genome-wide association studies (GWASs) have been used to identify superior alleles and determine their use in forward breeding, in the study of the available genotypic variability of such complex traits [46]. The objective of this research was to study the genetic regulation of drought tolerance at germination and during seedling growth in maize to identify associated genomic regions and candidate genes.

## 2. Results

### 2.1. Phenotypic Variation and Establishment

Drought stress had detrimental effects and the mean values of each trait were significantly reduced under water stress compared to control conditions (Table 1). The reduction caused by drought varied from 90% in coleoptile length and dry weight to 40% for germination. A large phenotypic variation for all traits was found under control conditions and under induced drought (Table 1). The ranges of variation were reduced under drought compared to control conditions, except for germination, and the minimum was zero for all traits under drought stress. The data obtained showed that the distribution of the traits was approximately symmetric for germination and root length under both conditions. The phenotypic values of all other traits were positively skewed in both conditions, except for germination (Table 1).

### 2.2. Phenotypic Correlations between Measured Traits

Phenotypic correlations between the genotypes under each treatment showed different responses to drought stress (Table 2). Under well-watered conditions, significant (*p* < 0.01) correlations were observed between germination (G%) and establishment related-traits. All correlations were positive except those between germination and the ratios coleoptile length/root length (CL/RL) and coleoptile dry weight/root dry weight (CDW/RDW). Coleoptile length (CL) was strongly correlated with root length (RL) (r = 0.75) and coleoptile dry weight (CDW) (r = 0.83). In addition, root length (RL) was highly correlated with coleoptile dry weight (CDW) (r = 0.70) and root dry weight (RDW) (r = 0.74), and the ratio coleoptile dry weight/root dry weight (CDW/RDW) was positively correlated with the ratio coleoptile length/root length (CL/RL) (r = 0.68).

Under water stress conditions, all correlations between germination (G%) and establishment related-traits were significant (*p* < 0.01) and positive, and germination (G%) was highly correlated with root dry weight (RDW) (r = 0.74). Coleoptile length (CL) was highly correlated with coleoptile dry weight (CDW) (r = 0.85), and with the ratios coleoptile length/root length (CL/RL) (r = 0.84) and coleoptile dry weight/root dry weight (CDW/RDW) (r = 0.84). Other strong positive correlations were observed between the ratios coleoptile dry weight/root dry weight (CDW/RDW) and coleoptile length/root length (CL/RL) (r = 0.90), and coleoptile dry weight (CDW) (r = 0.75).

Correlations differed between drought and control conditions, particularly between germination and CL/CR, germination and CDW/RDW, RDW and CL/RL, CDW/RDW and RL. Between CDW/RDW and RDW, the correlation was positive under drought and negative under control conditions. The correlation between CDW and CL/RL was significant under drought and not significant under control conditions. Conversely, between CL/RL and RL, correlation was significant under control and not significant under drought conditions.

### 2.3. Genomic Regions Associated with Germination and Seedling Establishment Related Traits

GWAS analysis of the 420 inbred lines with their eight parents used in the current study was performed to identify the genetic associations with the studied traits. Twenty-eight were significantly associated with seedling traits under water stress and 16 SNPs were significantly associated under well-watered conditions (*p* threshold = 2.42 × 10^−5^).

In this study, no SNPs were significantly associated with germination and root length under water stress. Three SNPs, distributed on chromosomes 5, 6, and 7, were significantly associated with coleoptile length. The additive effect associated with individual SNPs ranged between 0.76 and 1.62. The SNPs explained between 9 and 14% of the variance, and the number of homozygous lines with the unfavorable allele was higher than the number of lines with the favorable allele for coleoptile length, except for the QTL on bin 5.05 (Table 3).

Eight SNPs, distributed along the chromosomes 2, 4, 6, and 9, were significantly associated with the ratio of coleoptile length/root length under water stress. The additive effect ranged between 0.07 and 0.14 cm. The percentage of variance explained by individual SNPs ranged from 7 to 16%, and the SNP S9_10122782 reached the highest percentage. The number of homozygous lines with the allele that reduced the ratio was higher than the number of homozygous lines with the allele that increased the ratio (Table 3).

Ten SNPs distributed along the chromosomes 2, 5, 6, and 8 were significantly associated with coleoptile dry weight, with three in bin 2.04 and two in bin 8.06. The additive effect for individual SNPs ranged between 3.68 and 6.97 cm, and the SNPs explained between 7 and 16% of the variance, and the highest value was reached by S5_11084653. The number of homozygous lines with the unfavorable allele was higher than the number of homozygous lines with the favorable allele for the trait (Table 3).

The two SNPs, S3_10685613 and S10_127256724, on chromosome 3 and 10, respectively, were significantly associated with root dry weight under water stress, with an additive value of 2.68 and 3, respectively, and the percentage of variance was 10 and 11%. The number of homozygous lines with the unfavorable allele was higher than the number of homozygous lines with the favorable allele (Table 3).

Five SNPs distributed along chromosomes 1, 2, and 9, were significantly associated with coleoptile dry weight/root dry weight ratio under water stress, with three of them in bin 2.08. Three SNPs were found on chromosome 2. The additive value associated with individual SNPs ranged from 0.41 to 0.52 cm and the percentage of variance ranged between 10 and 15%. The number of homozygous lines with the unfavorable allele was higher than the number of homozygous lines with the favorable allele (Table 3).

Under well-watered conditions, no SNPs were found to be significantly associated with root length, root dry weight, and coleoptile dry weight/root dry weight ratio. Three SNPs distributed on the chromosomes 1 and 4 were significantly associated with germination in this treatment, and the SNPs S4_18695379 and S4_18695411 in bin 4.03 were close. The additive effect associated with individual SNPs ranged between 8.27 and 13.06 cm (shared between the two SNPs on chromosome 4), and the percentage of variance ranged from 12 to 15% (shared between the two SNPs on chromosome 4). The number of homozygous lines with the favorable allele was higher than the homozygous lines with the unfavorable allele (Table 4).

Only one SNP on the chromosome 6 was significantly associated with coleoptile length, with an additive value of 3.82, where the percentage of variance explained with this SNP was 12%. The number of homozygous lines with the favorable allele was lower than the number of lines with the unfavorable allele (Table 4).

Eleven SNPs distributed along the chromosomes, with the exception of chromosomes 4, 5, and 9, were significantly associated with the coleoptile length/root length ratio. The additive value ranged from 0.11 to 0.27 cm and the SNPs S8_171777809 had the higher value (0.27). The percentage of variance explained between 7 and 14%. The number of homozygous lines with the favorable allele was lower than the number of homozygous lines with the unfavorable allele (Table 4).

One SNP located on the chromosome 7, S7_140235159, was significantly associated with coleoptile dry weight. The additive value of this SNP was 0.36 cm and the percentage of variance explained by this SNP was 9%. The number of homozygous lines with the favorable allele was lower than the number of homozygous lines with the unfavorable allele (Table 4).

Some SNPs were associated with two traits; specifically, under drought stress, the SNP S9_10122782, located in bin 9.01 and associated with coleoptile dry weight/root dry weight ratio, was also associated with coleoptile length/root length ratio. The SNP S6_3045454 located in in bin 6.00 was associated with coleoptile length and was very close to S6_3042595, which was associated with coleoptile dry weight. Finally, under control conditions, S7_140235159, in bin 7.03, was associated with coleoptile length/root length ratio and coleoptile dry weight.

### 2.4. Candidate Genes Described in the Regions Surrounding Significant SNP

Based on the maize genome reference, a list of candidate genes was retrieved within an interval of 700 kbp upstream and downstream from the significant SNP. We considered as candidate genes those genes previously reported in the 700 kbp interval of each SNP significantly associated with a trait, which had some potential relationship with the trait associated with the significant SNP, based on the described function of the gene. There were 1566 candidate genes associated with all traits under well-watered (631) and stress (935) conditions. The most significant of these are reported in Appendix A.

Under well-watered conditions, there were 71 candidate genes associated with germination in chromosomes 1 and 4, and 67 candidate genes associated with coleoptile length in chromosome 6. Coleoptile length/root length ratio had the highest number (457) of candidate genes under well-watered conditions in chromosomes 1, 2, 3, 6, 7, 8, and 10. In addition, there were 36 candidate genes associated with coleoptile dry weight in chromosome 7 (Appendix A).

Under water stress conditions, 70 candidate genes were associated with coleoptile length in chromosomes 5, 6, and 7. There were 367 candidate genes for coleoptile dry weight in chromosomes 2, 5, 6, and 8. For coleoptile length/root length ratio, there were 261 candidate genes in chromosomes 2, 4, 6, and 9. There were 170 candidate genes associated with coleoptile dry weight/root dry weight ratio in chromosomes 1, 2, and 9. Finally, 67 candidate genes were associated with root dry weight in chromosomes 3 and 10 (Appendix A).

Among these candidate genes, only those related to drought stress, germination, and seedling traits are detailed in the Discussion section. The candidate genes detected under water stress and well-watered conditions involved in other stresses and biological processes are listed with their implications in Appendix A.

## 3. Discussion

Drought is a serious agronomic problem and one of the most important factors limiting maize biomass and seed production in almost all areas where it is grown. As a quantitative trait, it requires an understanding of genetic mechanisms controlling various plant responses for adopting different breeding approaches [29].

In this study, drought was simulated through the solution of Polyethylene glycol (PEG). PEG-induced drought decreases water uptake and thus germination, which becomes either delayed or occurs at a reduced rate [47]. In a previous study conducted by Djemel et al. [48], the applied stress (200 g/L of Polyethylene glycol) had the highest effect on germination indexes in the temperate maize germplasm [48].

The RILS used in the current study showed a wide range of phenotypic variation at the early developmental stage in response to drought-induced stress. The applied drought stress treatment clearly exerted a negative impact on germination and seedling performance of all maize inbred lines by retarding shoot and root-related traits compared to the control. This is in agreement with the observations of Ali et al. [49] and Khan et al. [42].

In previous studies, the MAGIC population used here allowed a finer dissection of the genetics of maize resistance to corn borers and a strong detection of candidate genes based on functional information [50,51,52,53,54].

The targeted phenotypic traits are considered important for the study of drought tolerance at seedling stage and in breeding programs. The germination percentage is the most representative trait at this stage, as noted by Grzesiak et al. [55]. The germination rate under drought stress can indicate the germination ability of drought resistance of various varieties and lines, and root length and density are good indicators of drought tolerance in cultivated crops [56]. The root–shoot ratio has been identified as an important trait indicator of drought tolerance in maize [57]. Avramova et al. [34] identified total root length and shoot dry weight as being reliable measurements of drought tolerance at the seedling stage under field conditions in maize.

These traits have been used in several studies for selection and screening under drought at the seedling stage in maize. The principal effect of a water deficit imposed by drought is impaired germination, resulting in poor plant stand at the early seedling phase and hampering early crop establishment [58]. Li et al. [36] and Naveed et al. [37] explained that, although both shoot and root growth were inhibited by drought stress, shoot growth was more sensitive than root growth. Thus, the shoot–root ratio was typically reduced, which is consistent with our results and those of Ruta et al. [39]. This implies that, under drought stress, plants allocate more resources to root development than to shoot development in order to enhance water acquisition and limit evaporation [59]. Liu et al. [30] reported that, from a physiological perspective, a high root–shoot ratio is, in general, beneficial to tolerate drought stress. Shao et al. [60] associated the reduction in seedling growth to a restricted cell division and enlargement, because drought stress directly reduces growth by decreasing cell division and elongation.

Our results revealed associations between germination and other seedling traits, through significant correlations obtained between germination and targeted traits. Radić et al. [61] noted that poor maize seed germination is directly associated with poor post germination performance. Aslam et al. [62] also confirmed this finding. These data show that drought stress has a clear impact on seed germination and growth and other performance-related traits in maize, although the detrimental effects varied among traits and there was notable variation among genotypes for response to drought.

In the present study, GWAS was carried out on maize inbred lines phenotyped for root and coleoptile-related traits under drought and well-watered conditions, in order to detect SNPs and genomic regions involved in drought tolerance, and thus identify candidate genes which underlie our traits of interest. Some SNPs showed clear pleiotropy with other traits, which is consistent with the significantly positive correlations between the phenotypes. Under stress conditions, the SNP S9_10122782 was significantly associated with the two traits CDW/RDW and CL/RL, which is consistent with the high and significantly positive correlations observed between these traits (r = 0.90). In addition, the SNP S6_3045454, associated with CL, was very close to S6_3042595, associated with CDW, and these traits were also highly correlated (r = 0.83). Under well-watered conditions, the SNP S7_140235159 was significantly associated with CDW/RDW and CL/RL, which is consistent with the significant and positive correlations between the traits (r = 0.68).

The density of SNPs detected in our work varied among chromosomes; the maximum number of SNPs were found on chromosome 6 (nine SNPs) and chromosome 2 (eight SNPs) for all the traits under well-watered and drought conditions. This may increase the interest in these genomic regions for future research. On chromosome 6, the SNP S6_111018551 associated with coleoptile length/root length ratio under drought stress conditions in our study, was detected in a QTL region in a previous study conducted by Trampe [63] using an F2:3 population derived from inbred A427 and CR1Ht. The QTL region spans markers S6_111018551–S6_111368312 at 82 cm, and the QTL showed a pleiotropic effect for anther emergence, pollen production, tassel size, and haploid male fertility.

On chromosome 2, the SNPs S2_55483944, S2_56924044, and S2_61029060 significantly associated with coleoptile dry weight under water stress conditions were located in the chromosomal region (Bin 2.04) where the root-ABA1 QTL is also located [64]. This exerts an important effect on L-abscisic acid and stomatal conductance, influences root lodging through a constitutive effect on root architecture [64], and affects grain yield [65].

Yi et al. [52] studied QTLs for germination and seedling traits in the same MAGIC population under control and cold conditions, and found QTLs for some traits close to those found in our current results. Under drought conditions, we found a QTL for coleoptile/root length ratio in bin 2.02, and Yi et al. [52] reported an important concentration of SNPs associated with early vigor, chlorophyll content, and efficiency of photosystem II under cold conditions. Bin 2.04 had several QTLs for efficiency of photosystem II under cold conditions close to the QTL in the current coleoptile dry weight. Bin 5.05 had an SNP associated with chlorophyll under cold conditions, at least one million databases from the QTLs associated with coleoptile length in the current study. Bin 6.04 had a QTL for chlorophyll under cold conditions close to the QTL for coleoptile/root length ratio found here. These similitudes could indicate common genetic regulation for early plant development under drought and cold stresses. The co-localizations were fewer under control conditions. We detected a QTL for coleoptile/root length in bin 3.05 and Yi et al. [52] reported two close QTLs for chlorophyll under control conditions. However, although Yi et al. [52] found hundreds of QTLs, no other reliable co-localizations were identified, particularly under control conditions.

Based on the maize genome reference, several candidate genes were identified close to the SNPs significantly associated with the traits. Among these candidates, there were those related to biotic and abiotic stresses, growth, plant development, and biological processes. Here, we highlight the genes involved in germination and seedling development under abiotic stresses, particularly drought stress. Others, with their implications, are noted in the Appendix A.

The candidate gene *Zm00001d035993*, associated with the ratio CL/RL, and *Zm00001d021163*, associated with CDW, under well-watered conditions, and *Zm00001d022079* associated with CL and *Zm00001d007251* associated with the ratio CDW/RDW under drought conditions, belong to the *nucleobase cation symporter 2 (NCS2)* gene family. NCS2 genes in maize are involved in diverse developmental processes and responses to abiotic stresses, including abscisic acid, salt (NaCl), Polyethylene glycol, and low (4 °C) and high (42 °C) temperatures. These genes are important for the transport of free nucleobases, participating in diverse plant growth and developmental processes, and response to abiotic stress [66].

Under well-watered conditions, the genes *Zm00001d038676* and *Zm00001d038695* were identified as candidates for coleoptile length in our study. *Zm00001d038676* was expressed in a drought tolerant maize mutant (*C7-2t*) implicated in cell wall enrichment conferring drought resistance [67]. In a study conducted by Wang et al. [68], the gene *Zm00001d038695* (*gibberellin 2-oxidase*) was up-regulated after heat stress; this is related to gibberellin acid (GA) biosynthesis, which increased after heat stress. Gibberellin acid hormones were involved in the heat stress response of young ears and had a significant adverse effect on their development.

Many candidate genes were associated with coleoptile length/root length ratio under well-watered conditions. In Li et al. [69], the gene *Zm00001d034129* (*Peroxidase 73*) was associated with maize seedling leaves under cold and heat stress. The authors reported that common genes for both treatments were enriched in the hydrogen peroxidase metabolic process, which is the case of this gene. A *MYB* family protein member *Zm00001d034160* is involved in modulating early nitrate responses in maize through the mechanism of alternative splicing (AS), which plays an important role in maize to adapt to nitrate fluctuation [70]. Zm00001d002707 was annotated as *cis-zeatin O-glucosyltransferase* with specificity to cis-zeatin, which contributes to the highly active cytokinin pool by inducing trans-isomers expression; this gene is highly similar to Evm.model.chr7.511, which exhibited a higher expression in the root of temperate lotus [71]. *Zm00001d012641* is a key gene involved in biotic and abiotic stresses, and is also related to zeatin biosynthesis, which was down-regulated under heat stress. In the study of Wang et al. [68], the endogenous hormone content, such as zeatin (ZT) in young ears, decreased significantly.

*Zm00001d042482* was annotated as *bHLH-transcription factor*. Numerous cellular processes and responses that are important for plants to tolerate various abiotic stresses are controlled by *bHLH*, which is a large family of conserved transcription factors [72]. The gene *Zm00001d021961* was annotated as *acetylcholinesterase1* (*ACHE1*). Native tropical zone plants show high acetylcholinesterase (*AChE*) activity during heat stress, and *AChE* activity in endodermal cells of maize seedlings is increased by heat treatment. Maize *AChE* is mainly expressed in coleoptile nodes and seeds. Enhanced maize *AChE* activity was observed after heat stress suggesting, that *AChE* plays a positive role in maize heat tolerance [73].

*Zm00001d008569* (CL/RL) (*Delta(7)-sterol-C5(6)-desaturase 1*) was a homolog of *Os01g0134500* in rice. This candidate gene is involved in brassinosteroid metabolism and was associated with various metaxylem vessel area-related phenotypes under well-watered conditions in maize and strongly associated with root metaxylem traits [74]. An impairment of its activity is linked to defective longitudinal growth, irregularly spaced vascular bundles, and reduced xylem vessel size and number [74].

A drought-responsive gene, *Zm00001d036003*, was detected under well-watered conditions and associated with the CL/RL ratio in our study. This gene belongs to the *ERF* family, genes of which are important regulators involved in cold and heat stresses. This gene was up-regulated under cold and heat conditions in Li et al. [69].

Under drought conditions, three genes involved in the ROS (Reactive Oxygen Species) scavenging system were detected. *Zm00001d017240*, which was associated with CL and is a *Grx-like* gene improving the stability of the plasma membrane biochemical process related to proline metabolism, was up-regulated in transgenic maize, enhancing drought tolerance. We conclude this gene may play an important role in ROS detoxification enzyme activity under drought stress [75]. *Zm00001d003797*, which was associated with CDW (*Ferredoxin-3 chloroplastic*), is a specific drought-responsive protein of the Sorghum bicolor genotype, RTx430 (preflowering drought-tolerant genotype), and is involved in the production of ROS scavenging ferredoxins, a mechanism of drought tolerance [76]. *Zm00001d037079*, which was associated with CL/RL, was down-regulated at the third leaf stage after waterlogging for 6 days [77].

The genes *Zm00001d017251* (*fatty aldehyde decarbonylase*) associated with CL and *Zm00001d039631* associated with RDW, implicated in suberine, wax, and cutin biosynthesis, were also detected, *Zm00001d017251* was up-regulated after drought stress during the juvenile phase of the maize plant growth. *Zm00001d039631* has a role in binding and transport of the fatty acid, cutin, and wax monomers, and was also up-regulated after 2 days of drought [78]. In genes involved in wax biosynthesis, the total amount of cuticular wax is increased in response to drought stress conditions, thus improving plant protection from water loss [78]. Relevant genes related to cell wall biosynthesis confirm the importance of the cell wall structure and composition in resistance to drought.

*Zm00001d017258* associated with CL was identified as an important *TCA* (*tricarboxylic acid*) gene overexpressed in *Arabidopsis* and associated with root development. The overexpression of this gene significantly shortened the length of the primary root of *Arabidopsis*, suggesting that elevated expression levels of these *TCA* cycle genes may cause certain negative effects on mitochondria and ultimately inhibit root growth [79].

The detected drought-responsive gene, *Zm00001d003850*, was associated with CDW (*Putative BOI-related E3 ubiquitin-protein ligase 2*). This is a specific gene of the tolerant line YE8112 in maize; it was up-regulated under water stress and encoding proteins involved in ubiquitination [80]. Protein ubiquitination has been widely recognized as a central regulator of stress responsive transcription factors and other regulatory proteins, effectively contributing to abiotic stress adaptation [81]. *Zm00001d002126* was associated with CL/RL and down-regulated in a drought-tolerant inbred line [80] *Zm00001d002199* receptor-like serine/threonine-protein kinase was identified as a hub-gene. Receptor kinases, another vital type of membrane protein, was found to be differentially expressed in response to drought stress in maize [82]. *Zm00001d037273* was associated with the CL/RL ratio and implicated in heat stress response (*HSR*). Jagtap et al. [83] detected *Zm00001d007215* in hybrids tolerant to water stress during the maize kernel-filling stage in a previous study. This is a differentially abundant proteins (*DAP*) gene involved in the response to drought stress [84]. *Zm00001d029707* was associated with CDW/RDW and is a *GST* gene induced and up-regulated after heat stress [70].

Zm00001d011406, which is a candidate for CDW, is a transcription factor belonging to the *HSFTF* (*Heat Shock Factor transcription factor*) family, was downregulated in response to ABA [85]. Heat shock transcription factor (*Hsf)* plays a transcriptional regulatory role in plants during heat stress and other abiotic stresses [86].

Two other candidate genes controlling CL/RL ratio were *Zm00001d002143*, which was associated with phenylpropanoid biosynthesis, a stress-responsive pathway [87], and *Zm00001d037112*, encoding a root-specific kinase protein, which was associated with abiotic stress responses in Rogers and Benfey [88].

*Zm00001d007229*, a candidate for CDW/RDW, belongs to the *PLC* (*Phospholipase C family*) gene family in maize. PLC is one of the main hydrolytic enzymes in the metabolism of phosphoinositide and plays an important role in a variety of signal transduction processes responding to plant growth, development, and stress [89]. The functional annotation of these genes in maize confirms the role of these candidates with regard to drought tolerance at the germination and early seedling development stage.

Genes involved in both heat and cold stresses were found to be associated with the CDW/RDW ratio. *Zm00001d007181* encoding CML (CaM-like proteins), was up-regulated under both cold and heat stress [70] and *Zm00001d007267*, which was mainly involved in photosynthesis, is a candidate gene identified as a possible hub gene involved in temperature stresses [69].

Genes implicated in waterlogging tolerance were detected. *Zm00001d022084*, which is a candidate for CL, was up-regulated under flooding conditions in maize. It was observed that the rice orthologue was also up-regulated under the same conditions, and this gene is implicated in flooding tolerance during seed germination and early seedling growth in rice [90]. *Zm00001d012103*, associated with CDW encoding of *aldolase2*, and involved in the glycolysis/gluconeogenesis pathway, which produces energy and recycles carbon for other pathways to survive, was found to be up-regulated, indicating this pathway’s underlying central role in waterlogging stress in early, late, and long-term responses [91]. *Zm00001d007161*, which was associated with the ratio CDW/RDW encoding peroxidase, is involved in response to waterlogging stress in maize roots [91].

Within the genes associated with our traits of interest under water stress, there were those implicated in salt, aluminum, and phosphorus stresses according to previous studies (Appendix A).

## 4. Materials and Methods

### 4.1. Plant Material

The maize multi-parent advanced generation inter-cross (MAGIC) population used in this study was established by the Maize Genetics and Breeding Group of the Misión Biológica de Galicia, CSIC (Pontevedra, Spain), with eight diverse founder lines, with a heterogeneous background [50,51] (Table 5). We evaluated 420 recombinant inbred lines (RILS) of the MAGIC population together with the eight parents for drought tolerance. The genotypes were phenotyped at germination and during seedling development, under control and water stress conditions imposed using aqueous solutions of high molecular weight Polyethylene glycol 6000 (PEG 6000) according to the method described by Álvarez-Iglesias et al. [41].

### 4.2. Experimental Design and Data Recorded

The evaluation of seed germination under water stress conditions was conducted in Petri dishes by placing 10 seeds of each maize recombinant inbred line on a Whatman filter paper layer moistened with 10 mL of PEG 6000 solution with a concentration of 200 g/L (considered to be a severe concentration for germination according to Alvarez-Iglesias et al. [41]), and distilled water was used as control. Petri dishes were sealed with parafilm to avoid water loss through evaporation and incubated in the dark at 27 °C in a growth chamber. The evaluation followed a randomized complete block design (RCBD) with three repetitions.

The number of germinated seeds was counted after 5 days and used to calculate the final germination percentage (G%). The primary root and coleoptile lengths (RL, CL) were recorded for all seedlings on each Petri dish. Then, the dry weight (DW) was obtained after drying for 72 h at 60 °C of all primary roots, secondary roots, and coleoptiles collected from each Petri dish. From these data, the ratios of coleoptile length/root length (CL/RL) and coleoptile dry weight/root dry weight (CDW/RDW) were calculated.

### 4.3. Statistical Analyses

The recombinant inbred lines derived from the MAGIC population were previously genotyped for 955.690 SNPs using a genotyping-by-sequencing (GBS) strategy. The genotype matrix was filtered as follows: Heterozygous genotypes and insertion/deletion polymorphisms (INDELs) were considered missing data and SNPs with more than 50% missing data and/or minor allele frequency less than 5% were omitted. As a result, we obtained a genotype matrix constituted by 215.131 SNPs distributed across the maize genome. The best linear unbiased estimators (BLUEs) for each trait were calculated. For each trait under both irrigation conditions, descriptive statistics (mean, skewness, kurtosis, and coefficient of variation) were calculated. Phenotypic (r_p_) correlation coefficients among traits under both irrigation conditions were calculated using REML estimates according to a SAS mixed model procedure [92]. A genome-wide association analysis was completed with Tassel 5 [93]. The association analysis was based on a mixed linear model (optimum compression level and P3D to estimate the variance components), which included a genotype-phenotype matrix and a kinship matrix built using the centered IBS method [94]. Complete and filtered genotype databases are available as supplementary materials in López-Malvar et al. [54].

### 4.4. SNPs, QTL and Candidate Gene Selection

To calculate the comparison-wise threshold for declaring an association between a trait and a SNP to be significant, we used a modification of the Bonferroni approach [95]. This consists of calculating the number of independent tests by the Haploview program using the option four gamete rules [96,97], resulting in 12,397 independent comparisons. Then, the comparison-wise threshold was the coefficient between the established experiment-wise threshold (0.3) and the number of independent tests. Thereby, a marker would be considered significantly associated with a trait at *p*-values less than 2.42 × 10^−5^ (−log10 (*p*-value) = 4.6).

A +/−700 kbp confidence interval region around each significant SNP was considered following previous association studies using the same MAGIC population [52,54]. SNPs were assigned to the same QTL in the case in which its confidence intervals overlapped. For the search of candidate genes associated with the traits under study, we considered the two described genes that delimit the +/−700 kbp region around the SNP in the reference genome assembly version 2. We positioned these in version 4 of the reference genome, and all genes contained in that region were then identified and characterized based on the maize B73 reference genome assembly (version 4) available on the MaizeGDB browser [98].

## 5. Conclusions

The present study shows how drought stress can affect seed germination and seedling parameters during early growth. The genomic regions and genes, and the metabolic pathways identified here, can be valuable genetic resources or selection targets for developing new drought-resistant maize cultivars. Some SNPs had significant associations with several traits, suggesting a pleiotropic genetic control. Other SNPs were co-located with QTLs for cold tolerance previously published for this MAGIC population. Several candidate genes, close to the significant SNPs, are related to biotic and abiotic stresses and plant development. These results can be used for improving drought tolerance at the seedling stage in maize in further research to increase the understanding of the genetic architecture of complex traits under drought stress conditions.

## Figures and Tables

**Table 1 plants-10-01786-t001:** Mean ± standard error (s.e.), coefficient of variation (CV), skewness and kurtosis, minimum and maximum of germination and establishment-related traits evaluated under water stress obtained with 200 g L^−1^ of Polyethylene glycol 6000 and control.

Variable	Mean ± s.e.	CV	Skewness	Kurtosis	Min.	Max.
*Control (0 g L^−1^ of Polyethylene glycol 6000)*						
Germination	74.44 ± 9.45	21.98	−0.77 **	0.38	20.0	100
Coleoptile length	21.34 ± 5.08	41.22	0.63 **	0.05	6.2	51.8
Root length	39.38 ± 8.75	38.46	0.50 **	−0.05	7.5	94.1
Coleoptile length/ Root length ratio	0.58 ± 0.14	41.43	2.99 **	16.85 **	0.2	2.5
Coleoptile dry weight	51.39 ± 16.18	54.53	1.07 **	1.29 **	6.7	160
Root dry weight	32.40 ± 11.64	62.15	1.77 **	4.98 **	3.3	136.7
Coleoptile dry weight/Root dry weight ratio	1.81 ± 0.44	42.16	1.54 **	3.06 **	0.7	5.2
*Water stress (200 g L^−1^ of Polyethylene glycol 6000)*						
Germination	42.95 ± 12.183	49.14	−0.02	−0.66 *	0	90
Coleoptile length	2.14 ± 1.343	108.73	1.34 **	1.76 **	0	11.6
Root length	15.95 ± 3.237	35.16	0.53 **	0.97 **	0	37.3
Coleoptile length/Root length ratio	0.13 ± 0.084	115.69	1.74 **	4.70 **	0	1
Coleoptile dry weight	5.42 ± 4.858	155.34	2.76 **	12.14 **	0	66.7
Root dry weight	10.68 ± 4.644	75.29	1.16 **	1.99 **	0	43.3
Coleoptile dry weight/Root dry weight ratio	0.46 ± 0.320	119.96	1.86 **	4.34 **	0	3.3

*, **, *** significant at 0.05, 0.01 and 0.001 probability level, respectively.

**Table 2 plants-10-01786-t002:** Phenotypic correlation coefficients among germination and establishment related-traits measured in the MAGIC population evaluated under water stress (below diagonal) obtained with 200 g L^−1^ of Polyethylene glycol 6000 and under well-water condition (above diagonal).

	G%	CL	RL	CL/RL	CDW	RDW	CDW/RDW
G%		0.25 **	0.37 **	−0.20 **	0.50 **	0.58 **	−0.26 **
CL	0.41 **		0.75 **	0.24 **	0.83 **	0.59 **	0.26 **
RL	0.47 **	0.40 **		−0.30 **	0.70 **	0.74 **	−0.19 **
CL/RL	0.29 **	0.84 **	0.07		0.10	−0.18 **	0.68 **
CDW	0.50 **	0.85 **	0.38 **	0.66 **		0.76 **	0.18 **
RDW	0.74 **	0.41 **	0.60 **	0.22 **	0.52 **		−0.34 **
CDW/RDW	0.26 **	0.84 **	0.14 *	0.90 **	0.75 **	0.17 **	

G% = germination, CL = coleoptile length, RL = root length, CL/RL = coleoptile length/root length ratio, CDW = coleoptile dry weight, RDW = root dry weight, CDW/RDW = coleoptile dry weight/root dry weight ratio. *, ** Significant at *p* = 0.05 and 0.01 respectively.

**Table 3 plants-10-01786-t003:** SNPs significantly associated with germination and seedling establishment related traits under water stress obtained with 200 g L^−1^ of Polyethylene glycol 6000. Chromosome position (bin), the number of QTLs, the significance of the association, the number of RILs with favorable and unfavorable alleles, the additive value, and the variance explained by each SNP are included in the table.

SNP^a^	Bin	QTL	*p* Value ^b^	Allele ^c^	Additive Effect ^d^	Increase/ Decrease ^e^	*R*² ^f^	*Previous Experiments with Co-Localizing QTLg*
*Coleoptile length*								
S5_185498400	5.05	CL5.05	1.92 × 10^−5^	C/T	0.76	97/87	0.1	
S6_3045454	6	CL6.00	2.29 × 10^−5^	A/G	1.19	20/184	0.09	
S7_163929841	7.04	CL7.04	2.17 × 10^−5^	A/G	1.62	11/128	0.14	
*Coleoptile length/Root length ratio*								
S2_6486062	2.02	CL/RL2.02	1.51 × 10^−5^	T/A	0.1	12/212	0.08	
S4_206750186	4.09	CL/RL4.09	7.41 × 10^−6^	C/A	0.13	7/135	0.13	
S4_235381511	4.09	CL/RL4.09	8.33 × 10^−6^	G/A	0.14	6/126	0.14	
S6_107737617	6.04	CL/RL6.04	2.97 × 10^−6^	T/C	0.1	14/234	0.09	
S6_111018551	6.04	1	1.27 × 10^−5^	A/G	0.09	14/250	0.08	Trampe (2019)
S6_114057282	6.04	CL/RL6.04	2.05 × 10^−5^	T/C	0.07	20/239	0.07	
S6_114919493	6.04	CL/RL6.04	2.11 × 10^−5^	T/C	0.08	20/164	0.09	
S9_10122782	9.01	CL/RL9.01	5.31 × 10^−7^	T/A	0.12	10/156	0.16	
*Coleoptile dry weight*								
S2_55483944	2.04	CDW2.04	6.01 × 10^−8^	G/C	6.97	12/230	0.13	
S2_56924044	2.04	CDW2.04	2.18 × 10^−5^	T/C	4.55	17/256	0.07	
S2_61029060	2.04	CDW2.04	1.44 × 10^−5^	A/C	5.28	13/241	0.08	
S5_11084653	5.02	CDW5.02	1.51 × 10^−5^	G/T	6.42	9/135	0.16	
S5_214081679	5.08	CDW5.08	7.67 × 10^−6^	T/A	5.93	12/122	0.13	
S6_3042595	6	CDW6.00	7.55 × 10^−7^	G/A	6.29	13/153	0.13	
S6_157412647	6.06	CDW6.06	9.23 × 10^−7^	T/G	7.12	10/127	0.15	
S8_145231325	8.05	CDW8.05	2.79 × 10^−6^	T/G	6.61	10/191	0.11	
S8_163741873	8.06	CDW8.06	4.82 × 10^−6^	C/T	3.68	25/174	0.11	
S8_164558505	8.06	CDW8.06	6.55 × 10^−7^	A/G	6.87	11/138	0.14	
*Root dry weight*								
S3_10685613	3.03	RDW3.03	1.18 × 10^−5^	A/C	2.68	73/116	0.1	
S10_127256724	10	RDW10.00	1.47 × 10^−5^	A/G	3	51/119	0.11	
*Coleoptile dry weight/Root dry weight ratio*								
S1_82154624	1.04	CDW/RDW1.04	1.48 × 10^−5^	G/A	0.52	6/127	0.14	
S2_216786049	2.08	CDW/RDW2.08	1.79 × 10^−5^	T/G	0.48	7/147	0.11	
S2_217507453	2.08	CDW/RDW2.08	6.87 × 10^−6^	T/A	0.52	7/118	0.15	
S2_218777697	2.08	CDW/RDW2.08	3.85 × 10^−6^	A/G	0.41	11/193	0.1	
S9_10122782	9.01	CDW/RDW9.01	1.48 × 10^−5^	T/A	0.41	10/153	0.12	

^a^ The number before the underscore (_) indicates the chromosome number and the number after the underscore (_) indicates the physical position of the SNP in bp within the chromosome. ^b^ The significance threshold based on the deviation of F observed from expected is *p* = 1 × 10^−4^. ^c^ The allele before the slash (/) increases the trait and the allele after the slash decreases the trait. ^d^ The additive effect was calculated as half the difference between the mean number of homozygotes for the allele that increases the trait and the mean number of the allele that decreases it. ^e^ Number of homozygous lines for a given variant. The number before the slash refers to the allele that increases the trait and the number after the slash to the allele that decreases the trait. ^f^
*R*^2^, proportion of the phenotypic variance explained by the SNP.

**Table 4 plants-10-01786-t004:** SNPs significantly associated with germination and seedling establishment-related traits under well-watered conditions. Chromosome position (bin), the number of QTLs, the significance of the association, the number of RILs with favorable and unfavorable alleles, the additive value, and the variance explained by each SNP are included in the table.

SNP ^a^	Bin	QTL	*p* Value ^b^	Allele ^c^	Additive Effect ^d^	Increase/ Decrease ^e^	*R*² ^f^	*Previous Experiments with Co-Localizing QTLg*
*Germination*								
S1_281709337	1.1	G1.1	6.63 × 10^−6^	A/T	8.27	116/31	0.15	
S4_18695379	4.03	G4.03	8.67 × 10^−7^	C/T	13.06	209/12	0.12	
S4_18695411	4.03	G4.03	8.67 × 10^−7^	C/G	13.06	209/12	0.12	
*Coleoptile length*								
S6_158689148	6.06	CL6.06	1.74 × 10^−5^	C/T	3.82	32/141	0.12	
*Coleoptile length/Root length ratio*								
S1_42298404	1.03	CL/RL1.03	4.38 × 10^−6^	A/C	0.19	9/165	0.1	
S1_279913636	1.1	CL/RL1.1	1.38 × 10^−5^	C/G	0.15	14/130	0.11	
S2_19259124	2.03	CL/RL2.03	1.28 × 10^−7^	G/A	0.22	10/197	0.12	
S3_165855897	3.05	CL/RL3.05	1.22 × 10^−5^	A/G	0.2	8/149	0.12	
S6_62741341	6.01	CL/RL6.01	2.29 × 10^−5^	G/C	0.11	32/218	0.12	
S7_140235159	7.03	CL/RL7.03	1.11 × 10^−6^	C/T	0.16	11/168	0.1	
S7_161684211	7.04	CL/RL7.04	1.40 × 10^−5^	A/G	0.13	23/109	0.07	
S8_13332626	8.02	CL/RL8.02	1.64 × 10^−5^	A/C	0.22	7/129	0.14	
S8_159549553	8.06	CL/RL8.06	8.90 × 10^−6^	A/T	0.12	20/232	0.12	
S8_171777809	8.08	CL/RL8.08	9.84 × 10^−6^	C/G	0.27	5/122	0.12	
S10_18140694	10.03	CL/RL10.03	5.14 × 10^−6^	A/C	0.19	9/165	0.13	
*Coleoptile dry weight*								
S7_140235159	7.03	CDW7.03	1.17 × 10^−5^	A/G	0.36	28/183	0.09	

^a^ The number before the underscore (_) indicates the chromosome number and the number after the underscore (_) indicates the physical position of the SNP in bp within the chromosome. ^b^ The significance threshold based on the deviation of F observed from expected is *p* = 1 × 10^−4^. ^c^ The allele before the slash (/) increases the trait and the allele after the slash decreases the trait. ^d^ The additive effect was calculated as half the difference between the mean number of homozygotes for the allele that increases the trait and the mean number of the allele that decreases it. ^e^ Obs = number of homozygous lines for a given variant. The number before the slash refers to the allele that increases the trait and the number after the slash to the allele that decreases the trait. ^f^
*R*^2^, proportion of the phenotypic variance explained by the SNP.

**Table 5 plants-10-01786-t005:** Parental lines of the MAGIC population evaluated in this study.

Lines	Grain Color	Pedigree	Type of Grain
EP17 ^a^	Yellow	A1267 (Unknown location) ^e^	Flint
EP43 ^a^	Yellow	Parderrubias (Atlantic Spain) ^e^	Flint
EP53 ^a^	Yellow	Laro (Atlantic Spain) ^e^	Flint
EP86 ^a^	Yellow	Nostrano dell’Isola (Italy) ^e^	Flint
PB130 ^b^	Yellow	Rojo Vinoso de Aragón (Mediterranean Spain) ^e^	Flint
F473 ^c^	White	Doré de Gomer (France) ^e^	Flint
EP125 ^a^	Yellow	Selection from CO125	Corn Belt Dent
A509 ^d^	Yellow	A78 × A109	Corn Belt Dent

^a^ From Misión Biológica de Galicia (Spain). ^b^ From Estacão Agraria de Braga (Portugal). ^c^ From Institut National de la Recherche Agronomique (France). ^d^ From University of Minnesota (USA). ^e^ European landrace.

## Data Availability

Data are available from the authors upon request.

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
