# Peer review of "Genetics of Germination and Seedling Traits under Drought Stress in a MAGIC Population of Maize"

_plants, 2021, doi:10.3390/plants10091786_

Round 1

Reviewer 1 Report

This manuscript by Rida et al., reports QTLs relating to drought resistance at germination and seedling growth in maize. The authors carried out GWAS by using MAGIC population consisting of 420 RILs. They found 28 and 16 SNPs associated with germination and seedling traits under stress and well-watered conditions, respectively. This study will provide important information for understanding the mechanism of germination and seedling traits under drought stress. I think that this paper is acceptable after a few minor revisions, shown as mentioned below.

Line 222: The authors found the candidate genes in the regions surrounding significant SNP. But I couldn’t find the detailed definition of those candidates, that is how they select those candidates. All ORFs within an interval of 700 kbp upstream and downstream from the significant SNP were considered as candidate regardless of th presence or absence of SNPs in gene CDS? or only the genes carrying SNPs corresponding missense or nonsense mutations were included?. It should better to be described in materials or results section. Such information should be also shown for each gene in the Supplementary table 1.

Line 495-496: The authors described that “MAGIC population were previously1 genotyped” in the materials section. The data source (database and/or reference) should be specified if it has been already published or available on the public database. If the genotype data has not yet been reported, the authors should report it in this manuscript as a supplementary data.

Author Response

Dear editor,

Thank you for this valuable review and for the positive comments of the reviewers. We have made the corrections and responded to their questions below:

Reviewer Report 1

Line 222: The authors found the candidate genes in the regions surrounding significant SNP. But I couldn’t find the detailed definition of those candidates, that is how they select those candidates. All ORFs within an interval of 700 kbp upstream and downstream from the significant SNP were considered as candidate regardless of th presence or absence of SNPs in gene CDS? or only the genes carrying SNPs corresponding missense or nonsense mutations were included?. It should better to be described in materials or results section. Such information should be also shown for each gene in the Supplementary table 1.

Response: we considered as candidate genes those genes previously reported in the 700 kbp interval of each SNP significantly associated with a trait, which had some potential relationship with the trait associated with the significant SNP, based on the described function of the gene. This information has been included in the text and supplementary table 1

Line 495-496: The authors described that “MAGIC population were previously1 genotyped” in the materials section. The data source (database and/or reference) should be specified if it has been already published or available on the public database. If the genotype data has not yet been reported, the authors should report it in this manuscript as a supplementary data.

R: The data is available as supplementary material in another published paper (López-Malvar, A., Butron, A., Malvar, R.A. et al. Association mapping for maize stover yield and saccharification efficiency using a multiparent advanced generation intercross (MAGIC) population. Sci Rep 11, 3425 (2021). https://doi.org/10.1038/s41598-021-83107-1) to what now we referred in the manuscript.

Reviewer 2 Report

The review of manuscript:

General comments:

In the submitted manuscript, the authors studied genetic regulation of drought tolerance at germination  and seedling growth in maize. A genome-wide association study was carried out to identify genomic regions associated with drought tolerance. GWAS analysies identified SNPs significantlyassociated with germination and seedling traits under stress and well-watered conditions. Among the SNPs detected, two SNPs had significant associations with several traits with high  positive correlations, suggesting a pleiotropic genetic control. Other SNPs were located in regions that harbored QTL previously detected. The description of genomic regions comprised several candidate genes related to stresses and plant development, among them many drought-responsive genes and  transcription factors implicated in germination, seedling traits and drought tolerance is very interesting.

  1. The genotypes were phenotyped at germination and seedling development, under control and water stress conditions - is such research carried out only up to the seedling stage sufficient?
  2. It would be worthwhile to lead the plants further and estimate the yield - because that is what we care about. How do the results of work compare to the yields?
  3. Do you think that the same changes would be in the natural conditions of the plant return - that is, in the field?
  4. Have the SNP data been experimentally verified or is it just the result of bioinformatics analyzes?
  5. Please use the standard nomenclature for gene and protein writing. In general, symbols for genes are italicized, whereas symbols for proteins are not italicized for example http://www.biosciencewriters.com/Guidelines-for-Formatting-Gene-and-Protein-Names.aspx

Author Response

Reviewer Report 2

  1. The genotypes were phenotyped at germination and seedling development, under control and water stress conditions - is such research carried out only up to the seedling stage sufficient?

R: No, here we just begin with this study of the genetic regulation of response to drought at germination, which is the main limiting factor of maize production under drought. Indeed we will continue with studying the genetic regulation of the response to drought at successive studies; however, given the magnitude of these studies, and the independent regulation of response to drought at diverse stages, we decided to publish this first study without waiting for further studies at successive stages.

  1. It would be worthwhile to lead the plants further and estimate the yield - because that is what we care about. How do the results of work compare to the yields?

R: Although we do not have data to respond to that question, based on previous experience, resistance to stress at germination is basic for subsequent stages of development, but we do not expect to find a clear relationship between genes involved in resistance to stress at germination and genes regulating resistance at further stages.

  1. Do you think that the same changes would be in the natural conditions of the plant return - that is, in the field?

R: Although we do not have data to respond to that question, based on previous experience, evaluations under controlled conditions provide information of general value about the response to the specific stressful factor, removing interactions with other factors. Nevertheless, further field studies are necessary for having real information about field performance for each environment or group of environments.

  1. Have the SNP data been experimentally verified or is it just the result of bioinformatics analyzes?

R: The SNP data not been experimentally verified; therefore, these are just the result of bioinformatics analyses that will be discussed in subsequent studies.

  1. Please use the standard nomenclature for gene and protein writing. In general, symbols for genes are italicized, whereas symbols for proteins are not italicized for example http://www.biosciencewriters.com/Guidelines-for-Formatting-Gene-and-Protein-Names.aspx

R: We have made that correction

Reviewer 3 Report

The manuscript “Genetics of germination and seedling traits under drought stress in a MAGIC population of maize” describes the use of multi-parent advance generation inter-cross population as a useful tool to study drought stress in two developmental phases of maize, germination and seedling establishment, that are relevant for the prediction of harvest quality. The main objective of the article, related to the study of the genetic regulation of drought tolerance was accomplished.   Associated genetic regions and candidate genes involved in drought tolerance were determined.

The results were clearly present in different tables and figures and were discussed extensively and the materials and methods were clearly explained.

Author Response

Dear Reviewer, 

thankyou for finding time for reviewing our paper and for your comments

Round 2

Reviewer 2 Report

No further comments.